# Nearshore Bathymetry Retrieval from Wave-Based Inversion for Video Imagery

Diogo Santos [1], Tiago Abreu [2,3,*], Paulo A. Silva [2,4], Fábio Santos [1,2] and Paulo Baptista [1,2]

1   Department of Geosciences, Campus de Santiago, University of Aveiro, 3810-193 Aveiro, Portugal; dmps@ua.pt (D.S.); fabioacsantos@ua.pt (F.S.); renato.baganha@ua.pt (P.B.)
2   CESAM—Centre for Environmental and Marine Studies, University of Aveiro, 3810-193 Aveiro, Portugal; psilva@ua.pt
3   Department of Civil Engineering, School of Engineering-Polytechnic Institute of Porto, 4249-015 Porto, Portugal
4   Department of Physics, Campus de Santiago, University of Aveiro, 3810-193 Aveiro, Portugal
*   Correspondence: taa@isep.ipp.pt

**Abstract:** A wavelet-based method for bathymetry retrieval using a sequence of static images of the surface wave field, as obtained from video imagery, is proposed. Synthetic images of the water surface are generated from a numerical Boussinesq type model simulating the propagation of irregular waves. The spectral analysis is used to retrieve both wave periods and wavelengths by evaluating the spectral peaks in the time and spatial domains, respectively. The water depths are estimated using the linear dispersion relation and the results are validated with the model's bathymetry. To verify the proposed methodology, 2D and 3D simulations considering effects of wave shoaling and refraction were performed for different sea conditions over different seafloors. The method's ability to reproduce the original bathymetry is shown to be robust in intermediate and shallow waters, being also validated with a real case with images obtained with a shore-based video station. The main improvements of the new method compared to the consideration of a single image, as often used in Satellite Derived Bathymetry, is that the use of successive images enables the consideration of different wave periods, improving depth estimations and not requiring the use of subdomains or filters. This image processing methodology shows very positive results to provide bathymetry maps for shallow marine environments and can be useful to monitor the nearshore with high time- and space-resolution at low cost.

**Keywords:** bathymetry retrieval; wavelength; wavelet analysis; image sequences

## 1. Introduction

The littoral is one of the most dynamic regions of the Earth. Numerous complex interactions of atmospheric, hydrodynamic and sedimentary processes cause sea bottom changes and detailed nearshore bathymetry is essential to understand environmental processes, assess threats and mitigate climate change effects. Nevertheless, the spatial and temporal resolutions of bathymetric coverages of littoral areas are poor. A constant up-date of the morphological changes is essential, but this is impractical using standard methodologies. Particularly, the nearshore bathymetry in high energetic coastal regions, where waves steepen and break and intense bottom changes occur, do not allow to make use of conventional in situ survey methods such as acoustic systems (e.g., single and multibeam echosounders and other geophysical equipment) due to energetic hydrodynamic conditions in some periods of the year and consequent logistical commitments [1,2].

Given the economic and social implications of short-term scale bathymetry changes in littoral areas (e.g., safe navigation purposes at port entrances, potential hazardous situations associated to changes and developments of the coastline), it is important to develop methodologies that can provide bathymetric updates where changes are most

frequent and faster [3,4]. Remote sensing, especially in wave-dominated areas, can be a good alternative to obtain bathymetric information when conventional surveys (traditional echo sounding measurements) are difficult to conduct. Although the bathymetry derived from imagery is estimated rather than directly measured, it is one of the most cost-effective technologies covering large and remote areas [5,6]. Several image-based remote sensing algorithms to infer shallow water bathymetry can be found in the literature, being generally divided in two different main approaches [7]: the radiative transfer of light into the water and its interaction with the seafloor (e.g., [8,9]), and based on wave characteristics (e.g., [10,11]) methods. Since we envisage to monitor morphologic changes associated to energetic hydrodynamic conditions, we focus on wave-based algorithms where the bathymetry can be inferred from the surface wave field, making use of inverse methods and linear wave theory. The nearshore swell wave pattern is detected from the images and, through adequate processing steps that make use of spectral techniques, the corresponding wavelengths are calculated, and the depths inferred, relying on the inversion of the wave number dispersion relationship [10–12].

Depending on image resolution and employed algorithms, the results involve differences in terms of accuracies, depth ranges, point spacings and temporal variations. Despite these constraints, the development of Satellite Derived Bathymetry (SDB) led the International Hydrographic Organization (IHO) to consider acceptable SDB for areas uncovered by reference hydrographic surveys [13]. Better satellite image resolution is now available, but the spatial resolution is usually of (O > 10 m) [5]. Therefore, SDB are not suitable to monitor, for example, the evolution of longshore bars or of rip channels that require higher resolution. Major challenges are associated with requirements to resolve very small spatial features, of O(<10 m), and to track changes at small time scales. The use of video remote sensing systems (e.g., monitoring installations) seem to offer excellent spatial temporal resolutions, in combination with cost-efficient long-term data acquisition [14–16]. Different techniques to derive bathymetry in the very nearshore from shore-based video systems have also been under active development. Most of them are based on the wave phase estimation and depth inversion through linear and non-linear wave theories [17–20]. The accuracy of video-derived bathymetry is generally estimated with a spatially varying accuracy of tens of centimeters (e.g., [21,22]). By using a modified version of the cBathy spectral method [19] that allows to obtain depths from the wave number retrieval from video imagery, Bergsma et al. [23] evidenced the capability of near-continuous monitoring for 1.5 years. Their long-term video-derived bathymetry dataset demonstrates the possibility to continuously monitor the evolution of the beach morphology, allowing to make interesting morphological analyses with distinctive temporal resolution. Using a cross-correlation temporal method for depth inversion, Thuan et al. [24] provided three-year time-series of a beach profile evolution from video cameras. The authors also present guidelines on the limits of video-based depth inversion and for error assessment, which is crucial for operational systems.

Recently, Santos et al. [25] developed an image processing methodology using wavelet spectral analysis for the detection of the wavelength variation to map shallow marine environments. From the analysis of satellite images with a pixel resolution of 10 m, Santos et al. [26] derived bathymetry along 220 km of the high energetic Portuguese west coast and demonstrated the wavelet method extends the depth inversion limits of FFT towards shallower depths. This study aims to extend this conceptual wavelet methodology to a sequence of static images of the surface wave field, as obtained from video imagery. The method is based on a depth inversion technique, where both wave periods and wavelength variations along the wave propagation are identified from spectral analysis. Firstly, simulations of propagating waves are obtained from a numerical Boussinesq-type model that provides the sequence of images. The simulations are carried over different bottoms, with 2D and 3D features such as longshore bars or headlands. The inferred depths are validated with the real bathymetry, providing a comprehensive assessment of this methodology to reproduce main morphological features of shallow areas. The proposed methodology is

then applied to a set of shore-based video system images and the estimated bathymetry is compared with measurements.

## 2. Methodology

### 2.1. Depth Estimates

The use of remote sensing to estimate bathymetries in the nearshore can be based on physical wave characteristic changes (celerity, wavelength, wave height), due to the interaction with the seabed [27]. Waves propagating in intermediate and shallow waters show sea surface patterns (e.g., wavelength) directly related to the local bathymetry and can be detected with remote sensing. The wave crests and troughs present different characteristics in terms of pixel intensity values, allowing to retrieve bathymetric information through spectral analysis.

In this work, a wavelet spectral analysis technique is used for bathymetry retrieval. The wavelet transform (WT) allows decomposition of a signal into several parts to analyze them separately. In this way, it is possible to carry out a frequency analysis along the spatial/temporal domain, getting to know exactly when or where each of the frequency components occurs [28]. When applying the WT to a signal, an energy spectrum is obtained where the dominant frequencies correspond to the most energetic region of the energy spectrum.

Figure 1 shows an example of an energy spectrum of a periodic signal, $\eta$, resulting from the application of the Morlet wavelet used in this work. In a spatial domain of 4000 m, the signal contains a variation in its wavelength, $\lambda$, from left to right, changing almost linearly from 100 m to 300 m. The energy spectrum evidences the wavelength increase along the entire spatial domain, revealing that the application of the WT correctly estimates the variations of $\lambda$.

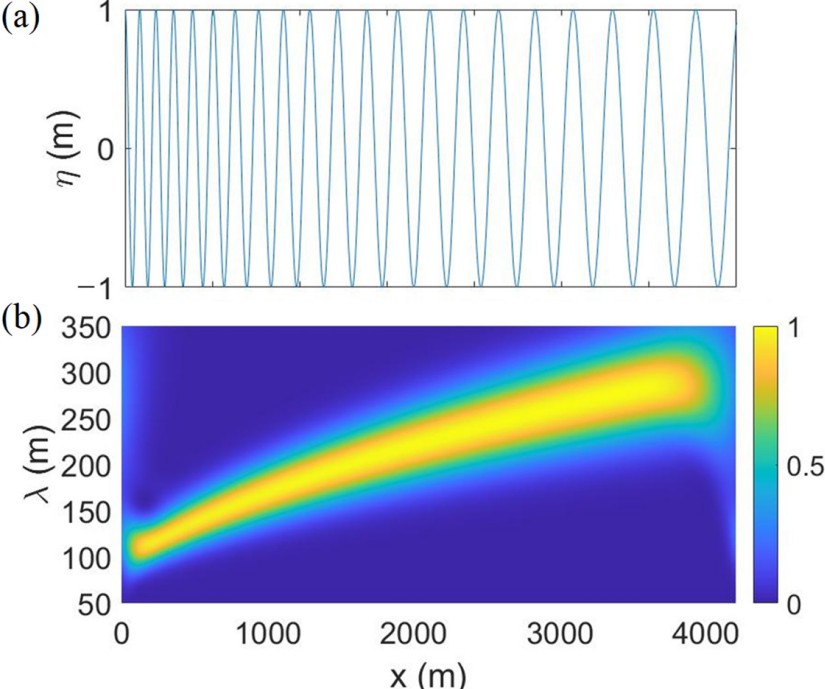

**Figure 1.** (**a**) Periodic signal and (**b**) corresponding wavelet energy spectrum.

There is a strong correlation between the local water depth of the seabed, $h$, and the wavelength of propagating waves via the dispersion relation

$$h = \frac{\lambda}{2\pi} atanh(\frac{\lambda}{\lambda_0}), \tag{1}$$

where $\lambda_0$ represents the wavelength at deep waters, which depends on the gravitational acceleration, $g$, and the wave period, $T$:

$$\lambda_0 = \frac{gT^2}{2\pi}. \tag{2}$$

Equation (1) represents the analytical solution of the dispersion relation of the Airy wave theory (often referred to as linear wave theory) that gives a linearized description of the propagation of gravity waves on the surface of a homogeneous fluid layer. In previous works (e.g., [11]) the value of $\lambda_0$ is retrieved from the analysis of sea surface imagery (e.g., SAR images) at offshore locations in deep waters and assumes a constant value in the dispersion relation. In this work, where a set of sequential images is considered, the wave period is estimated along time by the WT in the same way as the wavelength, and the ability to solve Equation (1) with constant or variable values of $T$ is assessed. This is one of the major differences regarding Santos et al.'s [25] methodology that considers a single image to retrieve bathymetric data and, as for most satellite techniques based on wave characteristics, adopts a constant offshore wave frequency.

*2.2. FUNWAVE Model*

Our goal is to apply the WT to a set of sequential static images of the surface wave field, as obtained from video imagery by remote sensing. Often, in real-life dynamic seabed situations, at the time of image capture the bottom is rarely known. Therefore, to validate the results of the proposed methodology, synthetic images were generated using a wave propagation model that replicates the temporal and spatial variations of the waves over different known bottoms. Furthermore, it is possible to simulate the propagation of irregular waves with different characteristics, identifying the best scenarios to retrieve nearshore bathymetric data from real images.

As the swell patterns in the ocean are associated to irregular waves, corresponding to the sum of several independent regular waves, there is a whole set of complex hydrodynamic processes related to nonlinear wave effects in shallow waters. Therefore, a numerical model that solves the Boussinesq equations is useful to simulate the wave propagation with different characteristics over different bathymetries. The use of numerical models that solve these equations has been greatly improved over time due to improvements in Boussinesq's theory, which benefits the use of this type of numerical models [29]. This study uses the FUNWAVE-TVD model developed by Kirby et al. [30], which is a numerical wave propagation model that solves Boussinesq equations. New versions were made available to improve and correct previous versions and these types of models have been widely used by the scientific community to simulate the propagation of waves in coastal areas. This model is based on the fully nonlinear Boussinesq equations proposed by Wei et al. [31], which implies that the results of this model can only be validated for intermediate and shallow waters.

Currently, the FUNWAVE model solves the equations using a hybrid method that combines finite volume schemes with finite difference schemes. This method has been used by several wave models of the Boussinesq type as it has shown a robust performance in wave propagation towards coastal zones [32]. More details of the numerical model can be found in Shi et al. [33].

*2.3. Numerical Simulations*

FUNWAVE was used to produce surface wave patterns in 2D and 3D domains where different scenarios were tested to assess the ability of the WT to estimate different bathymetries.

The 2D simulations consider three observed beach cross-shore profiles at Mira, Baleal and Carvalhido, which are located in the west coast of Portugal (Figure 2). All beaches exhibit submerged bars, but show different average slopes (5.2 8.6, 15.3 m/km, respectively). This data was obtained by the Portuguese COaStal Monitoring Program—COSMO—at

https://cosmo.apambiente.pt/ (accessed on 18 October 2021), which was conceived and developed by the Portuguese Environment Agency. This program consists of the collection, processing, and analysis of information on the evolution of beaches, dunes, nearby submarine bottoms, and cliffs along the mainland coast of Portugal. It is noted that the local depth measurements were taken up to about 22 m with multibeam echosounder. To extend the spatial domain to 3 km for the numerical simulations of Carvalhido steeper cross-shore profile, despite some bedform oscillations and since the average slope is fairly constant, interpolated values of the bathymetry were adopted for deeper depths, assuming the average slope of 15.3 m/km. Mira beach presents a gentler slope (5.2 m/km), evidencing a clear longshore bar and trough system. The submerged longshore bar extends for about 500 m in length and the location of the crest is located at about $x = 500$ m. Baleal cross-shore profile also exhibits submerged longshore bars and bedform oscillations close to the shore but with much smaller extensions.

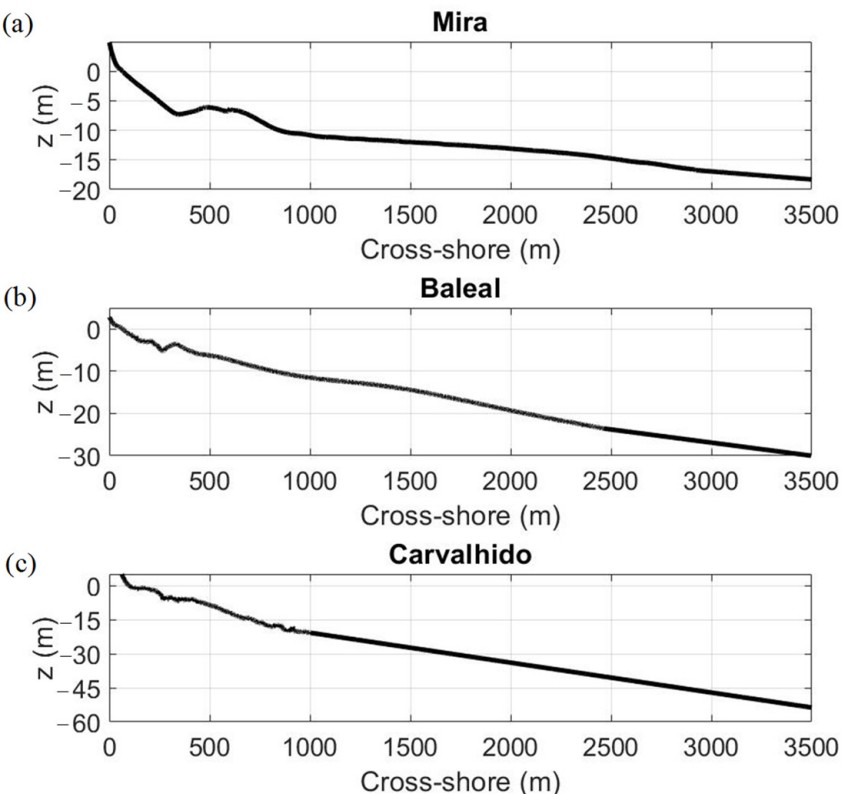

**Figure 2.** (**a**) Beach cross-shore profiles: (**a**) Mira, (**b**) Baleal and (**c**) Carvalhido.

The simulations were carried with a JON_1D wavemaker and the length of the bathymetric profiles was 3500 m. This wave generation method is based on JONSWAP formulation and is widely accepted and commonly used by the scientific community [34]. The peak enhancement factor, $\Upsilon$, adopted was 3.3. Since the simulations were carried out on transverse profiles perpendicular to the coast, the waves propagate at normal incidence.

Figure 3 shows a 3D spatial domain where a virtually headland was generated, corresponding to a $N \times M$ bathymetric profile ($N = 1000$ m and $M = 1500$ m). The waves were generated using a wavemaker (JON_2D), a method like JON_1D. In these simulations, also considering $\Upsilon = 3.3$, an incidence angle of 15° was used to assess if the methodology was able to correctly retrieve the bathymetry in the presence of refraction effects.

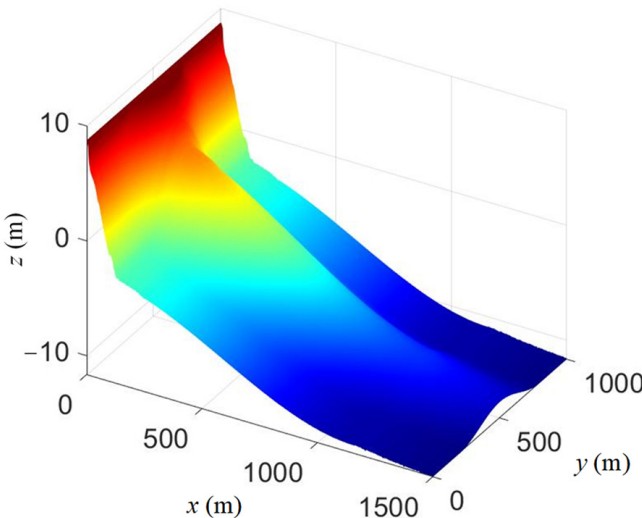

**Figure 3.** Headland dimensions.

In all simulations, a computational simulation time of 30 min was adopted, with the results being recorded every second ($\Delta t$ = 1 s). The grid spacing ($\Delta x$) is 1 m and the elevation $z$ = 0 refers to the mean sea level. The values of the model parameters related to wave propagation that are not indicated in this work were defined as default according to Shi et al. [29].

## 3. Results

### 3.1. Dispersion Relation: Influence of $\lambda_0$

As waves propagate towards the coast, they present variations in their characteristics (wave height, celerity, wavelength) due to the interaction with the seabed. To analyze the propagation of irregular waves over Mira beach cross-shore profile, simulations were carried with a significant wave height of $H_s$ = 1 m, a peak wave period of $T_p$ = 10 s and with a frequency range between 0.033 Hz to 0.33 Hz. After extracting each obtained 1D surface elevation pattern from FUNWAVE, a timestack image was generated for a time interval of 1200 s (Figure 4), corresponding to 20 min data records. The figure evidences the propagation of the wave crests over time making the wave trajectories visible and allowing to recover both wavelengths (along a line for a specific time) and wave periods (for each x in time) from spectral analysis. For each time, Equation (1) is solved to derive the estimated cross shore beach profile. To calculate a single depth result for each cross-shore position, the median statistical estimator was applied to the 1200 ensemble of solutions obtained in the temporal domain. The median is a robust statistical measure of central tendency capable of removing outliers, allowing automatically to extract anomalous wavelength estimations at each position.

By applying the methodology with a constant $T_p$ = 10 s, the inferred bathymetry is shown in Figure 5. The results obtained show large discrepancies of the depth estimations. Furthermore, for $x$ > 1.5 km, unexpected oscillations appear, accentuating the differences in relation to the real bottom profile.

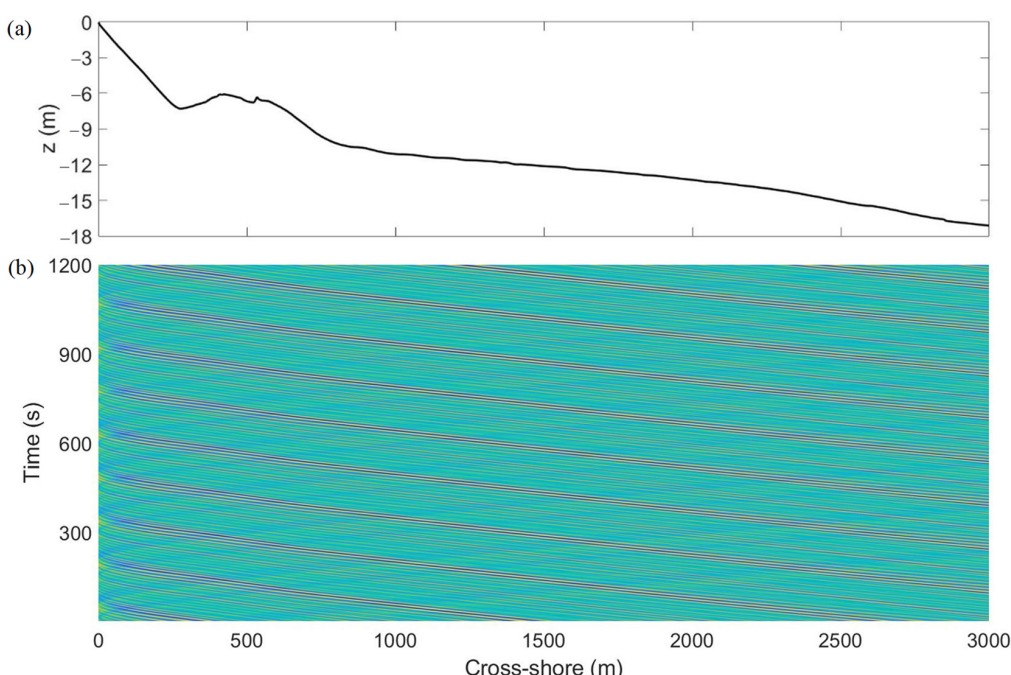

**Figure 4.** (**a**) Mira Beach cross-shore profile; (**b**) corresponding timestack, using FUNWAVE numerical simulations performed with $H_s$ = 1 m and $T_p$ = 10 s.

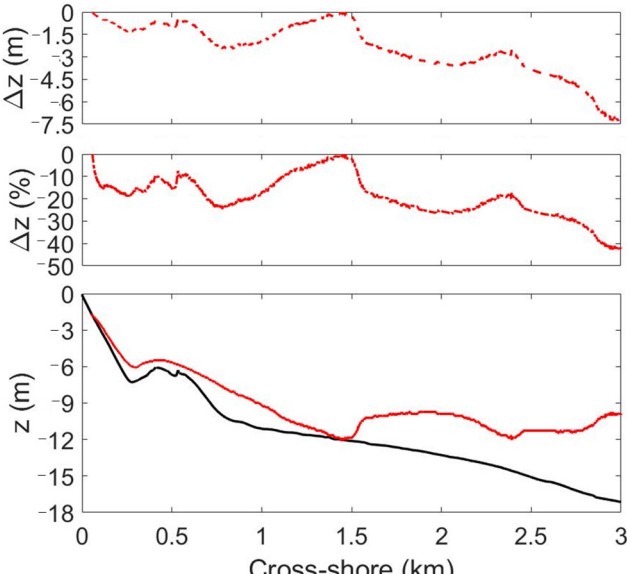

**Figure 5.** Cross-shore profile at Mira beach (black line) versus estimated water depths and respective errors and relative errors using constant $\lambda_0$ (red line and red dotted line).

A detailed analysis of the results for the wavelength estimates revealed the WT correctly computes the wavelength variations. Consequently, the errors observed in Figure 5 must be related to the application of the linear theory dispersion relation by considering a constant value in the computation of $\lambda_0$. Figure 6 represents the wave period and wavelength variation over time for $x$ = 1000 m by applying the WT. Analyzing the estimates of the wave period and wavelength at that location, it is observed that there is a very similar pattern between the oscillations of the two variables in the course of time. The wave period changes over time between $T$ = 6 s and $T$ = 14 s, oscillating around the expected average value of $T$ = 10 s. Therefore, the presence of this range of values of the irregular propagating

waves can imply that the depth estimates are being miscalculated when applying a constant value of $T = 10$ s.

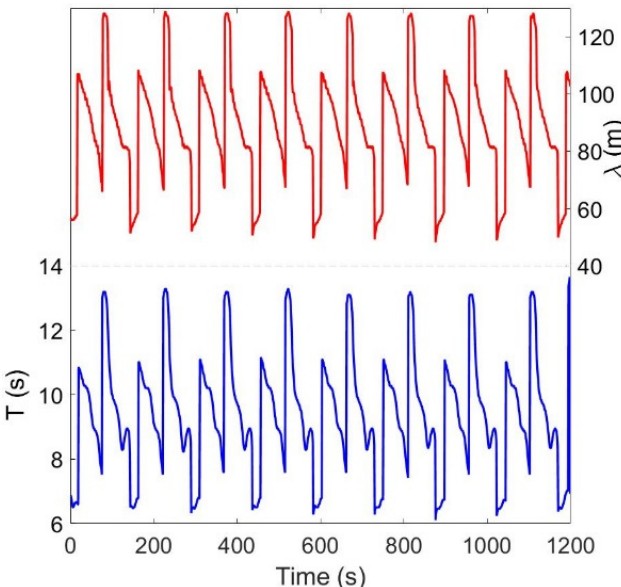

**Figure 6.** Wavelength (red) and wave period (blue) variations over time for $x = 1000$ m.

Figure 7 presents the results obtained from the dispersion relation, considering both wavelengths and wave periods variations. The results are clearly improved, corroborating the influence of T in the depth estimates. Along the spatial domain, the errors increase for greater water depths with mean absolute errors of 0.25 m up to $x = 2$ km and 0.67 m between $x = 2$ km and $x = 3$ km. This represents relative mean errors always below 5%, denoting an almost complete overlap of the two bathymetries. Noteworthy is the ability to detect the longshore bar with high precision, something that was not possible using the dispersion relation with a constant value of $T$.

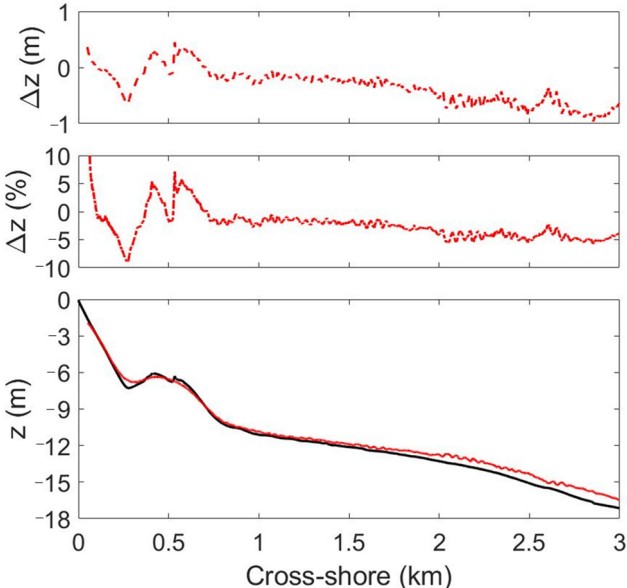

**Figure 7.** Real cross-shore profile at Mira beach (black line) versus estimated water depths and respective errors and relative errors using variable $T$ (red line and red dotted line).

### 3.2. Slope Influence

The bathymetric inversion methodology of Santos et al. [25] does not accurately estimate bathymetric profiles with steep slopes [26]. To analyze the performance of the proposed methodology based on a sequence of images, wave propagation simulations were carried over the sharp bathymetric profile of Carvalhido beach. Three swell patterns were evaluated: (i) $T_p = 6$ s with a frequency range between 0.05 Hz to 0.5 Hz; (ii) $T_p = 10$ s with a frequency range between 0.033 Hz to 0.33 Hz; and (iii) $T_p = 14$ s with a frequency range between 0.02 Hz to 0.2 Hz.

The depth estimates are shown in Figure 8 evidencing the consideration of different wave periods. The results show that as the wave period increases, the ability to reproduce greater depths increases. For $T_p = 6$ s, the depth estimates show small scale errors up to about 0.5 km where the depth is about 12 m. For $T_p = 10$ s it is possible to extend the calculation domain up to 1.3 km where the depth is about 26 m, and for $T_p = 14$ s it is possible to estimate the depth with minor errors up to $x = 2.5$ km where the depth is 41 m. In brief, the estimated depths in intermediate and shallow waters practically overlap the real bathymetric profile. This observation is in line with expectations since, for deeper depths, the interaction between the wave and the seabed is small and wavelength variations are not expected.

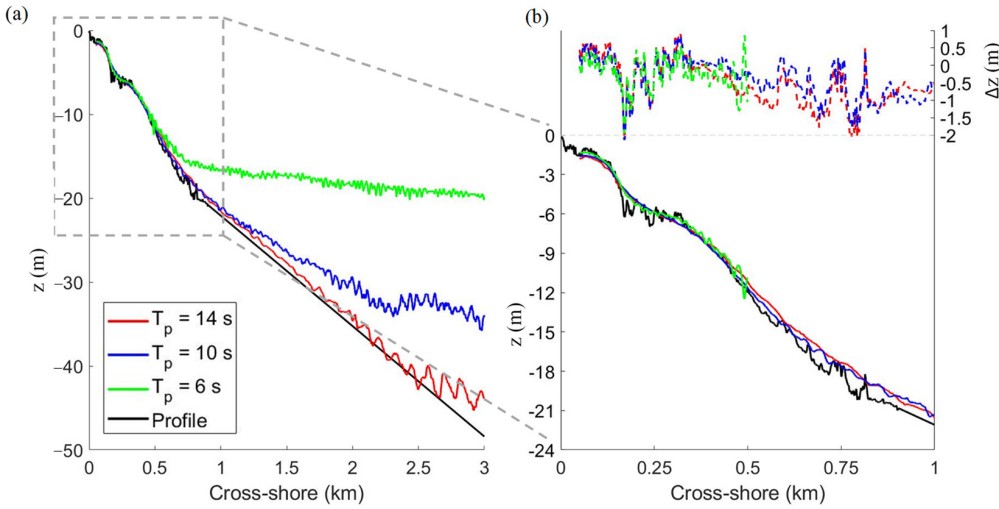

**Figure 8.** (**a**) Real cross-shore profile at Carvalhido beach (black line) versus estimated water depths using $T_p = 6$ s (green), $T_p = 10$ s (blue) and $T_p = 14$ s (red). (**b**) Details along the first kilometer show the respective estimation errors.

In general, due to the present seabed irregularities in shallow depths, there are relatively larger errors than those obtained for Mira beach. Figure 8b highlights the differences by zooming the previous results to the first kilometer of Carvalhido beach, evidencing errors always below 2 m. It is verified that there is a good approximation between all depth estimations and the profile used in the model, mainly in the first 500 m. The errors tend to increase when there are sudden variations in the bed profile, as for example, between $x = 0.15$ km and $x = 0.25$ km. The estimates appear to smooth out sudden variations in the beach profile and average errors obtained for the first 1000 m are 0.36 m, 0.54 m and 0.68 m, respectively, for $T_p = 6$ s, $T_p = 10$ s and $T_p = 14$ s. Despite these small errors, the results clearly show that this methodology allows to retrieve bathymetric data for steeper slopes with a high degree of precision, overcoming the limitations of Santos et al. [25] by using a single image.

### 3.3. Wave Period Influence on the Detection of Longshore Bars

Mira beach results show that the use of waves with $T_p = 10$ s allow to obtain good depth estimates enabling a good longshore bar description. The characteristics of the longshore

bars can change in terms of extension and elevation and multiple sandbar systems can appear along the spatial domain. To analyze the ability to detect smaller morphological variations, wave propagation over the Baleal beach was considered. In this case, narrow longshore bars appear close to the coastline.

Figure 9 shows the results by using three swell patterns with the previous hydrodynamic forcing ($T_p$ = 6 s, $T_p$ = 10 s and $T_p$ = 14 s). As in the case of the former analysis, as the wave period increases, the ability to obtain better estimations for greater depths increases. As expected for greater water depths, the bathymetric inversion fails. For this bathymetric profile presenting a smoother slope than Carvalhido beach, with $T_p$ = 6 s, the depth estimates present small-scale errors (error less than 1 m) up to about 1.25 km, where the depth is approximately 12 m. For $T_p$ = 10 s it is possible to extend good depth estimations up to 2 km, where the depth is about 20 m, and for $T_p$ = 14 s it is possible to estimate the depth with small errors throughout the entire spatial domain.

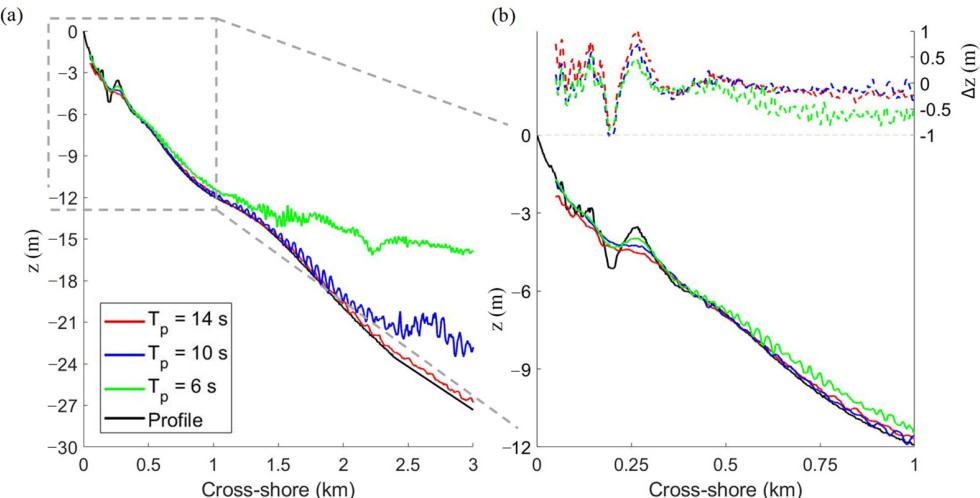

**Figure 9.** (**a**) Real cross-shore profile at Baleal beach (black line) versus estimated water depths and respective errors using $T_p$ = 6 s (green), $T_p$ = 10 s (blue) and $T_p$ = 14 s (red). (**b**) Details along the first kilometer and showing the respective estimation errors.

By zooming in the previous results for shallower depths corresponding to $x$ < 1 km (Figure 9b), it is verified that there is a good approximation of the depth estimations. The errors are always bellow 1 m, but larger fluctuations are observed for the different $T_p$, particularly at the location of the bars. Regarding the representation of bed forms such as the longshore bar at 0.25 km, smaller wave periods tend to have greater ability to represent this narrow bar. In the first 350 m, where there are sudden variations in depth, the average errors obtained for $T_p$ = 6 s, $T_p$ = 10 s and $T_p$ = 14 s were, respectively, 0.23 m, 0.31 m and 0.57 m. These errors evidence that waves with shorter $T_p$ and, therefore, shorter wavelengths allow better estimation of small variations in the nearshore morphology. Conversely, between $x$ = 350 m and $x$ = 1000 m, for $T_p$ = 6 s the errors approach 0.42 m while for $T_p$ = 10 s and $T_p$ = 14 s, the obtained errors decrease to about 0.11 m and 0.15 m, respectively. In general, for small wave periods, the surface waves can better adapt to the bottom reproducing smaller morphological variations present in shallow depths. However, it is necessary to consider that small wave periods only allow detecting with high precision a small part of the cross-shore profile.

### 3.4. Wave Refraction Influence on Depth Estimation

In the previous case studies, the waves approached perpendicular to the coastline (incidence angle of 0°). In the ocean, normally the waves hit the coast with an angle of incidence, $\theta$, making the waves to slowly change their direction due to refraction. This process makes wave crests bend according to water depth variations, driving them to become

perpendicular to the bathymetric gradient and inducing wavelength changes. To verify if the proposed methodology was able to correctly represent complex 3D bathymetries with refraction effects, a beach with a headland was considered in the numerical simulations. The simulations were carried with $H_s$ = 1 m, $\theta$ = 15° and $T_p$ = 10 s with a frequency range between 0.033 Hz to 0.33 Hz.

Figure 10 shows the bathymetric estimations as well as the differences in relation to the bathymetry introduced in the model. The results reveal an interesting agreement, showing that the simulated isobaths contours present a very similar pattern in relation to what is expected. Small differences can be observed but, in general, the bathymetry was reproduced with errors of less than 1 m throughout the entire spatial domain (average absolute error of 0.45 m). The biggest differences are found in shallow depths of the surf zone, where the validity of the linear dispersion relation is questionable [35,36]. Nevertheless, the application of the methodology enables to satisfactory retrieve good bathymetry data when refraction effects are present.

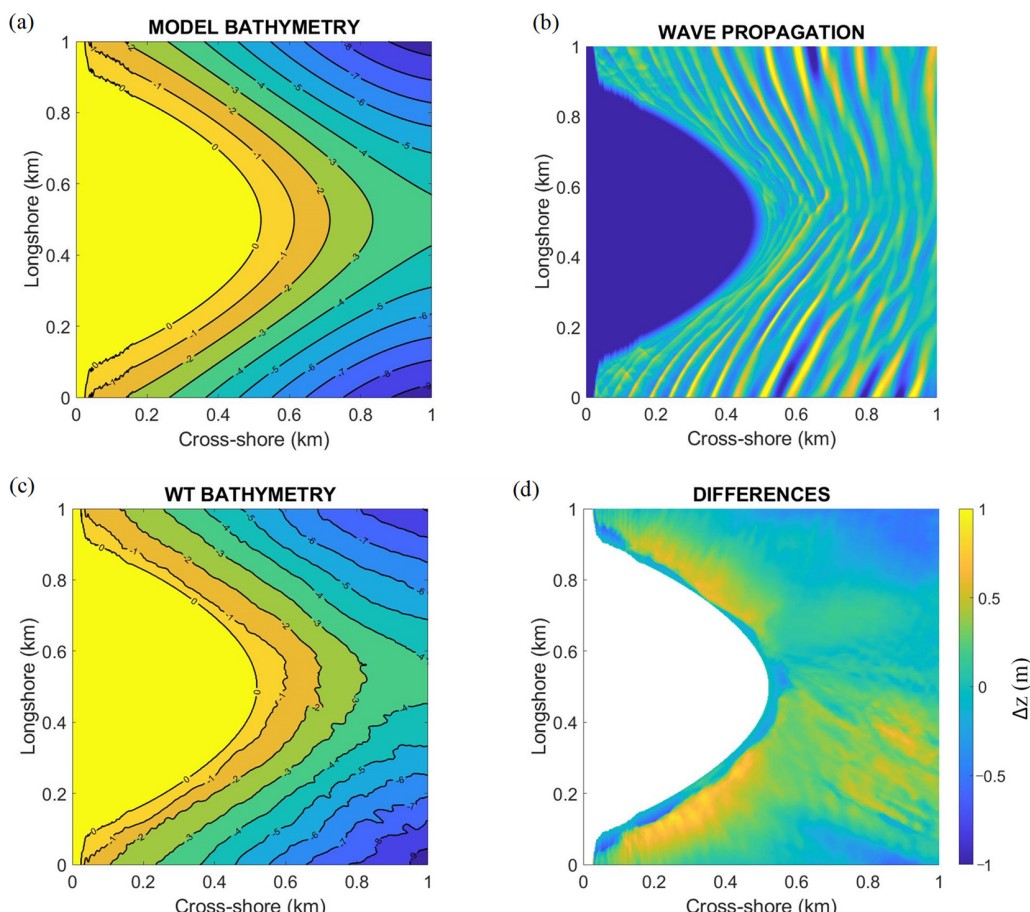

**Figure 10.** (**a**) Imposed bathymetry, (**b**) wave propagation, (**c**) estimated bathymetry and (**d**) differences between both bathymetries.

### 3.5. Case Study—Figueira da Foz

The proposed algorithm was tested with real data at the inlet of Figueira da Foz harbor (FFH) located in the west coast of Portugal. This is a wave dominated ebb-tidal delta with a submerged sand bar that constrains or endangers the navigation to the port. The sandbar is a persistent shallow feature which is quickly reestablished after dredging due to the intense longshore drift (Figure 11). Consequently, high frequency bathymetric surveys are carried out by the port authorities. A recent shore-based video station was installed at this location. The system captures video images with a 4K camera (Vivoteck IP9191-HP, New Taipei City, Taiwan) installed at the Sweet Atlantic Hotel and Spa building (40.15058°;

−8.86633°) at 70 m height and 1.5 km away from the study site (Figure 11). The camera was calibrated to eliminate distortions caused by the curvature of the lens and the video frames were rectified to transform oblique images into image-products typically projected on a horizontal plane (e.g., [15]).

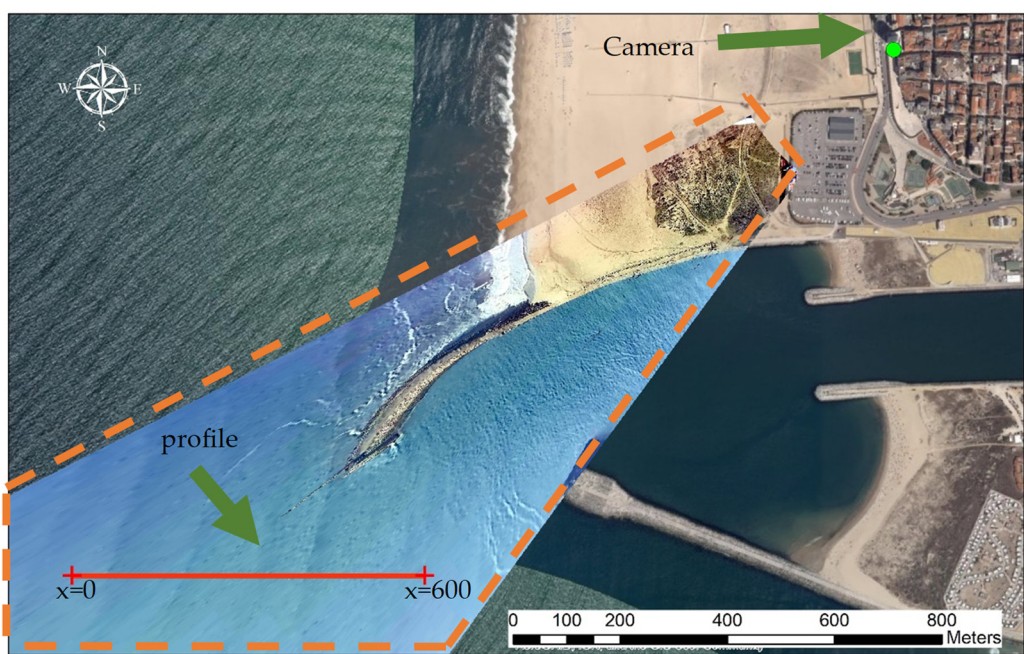

**Figure 11.** Figueira da Foz inlet. Location of the profile where the Timestacks are generated over a calibrated and rectified image. The orange-lined area shows the area covered by the video camera.

Video images for two consecutive days were selected (1 December 2020 and 2 December 2020), corresponding to a multibeam survey performed on 2 December 2020. During the analyzed period, the wave climate was very similar. On 1 December 2020 the wave period and wave height were, respectively, $T_p$ = 10 s and $H_s$ = 1.3 m, while on 2 December 2020 they were $T_p$ = 9 s and $H_s$ = 1.2 m. The video images considered for the analysis report to periods close to the slack high tide, where no breaking waves were observed in the area of interest and tide-induced currents were minor, and with no reflection of the sun on the water. The Timestack images for the chosen periods were generated for the profile indicated in Figure 11 with a spatial resolution of $\Delta x$ = 1 m and a temporal resolution of $\Delta t$ = 0.5 s (Figure 12).

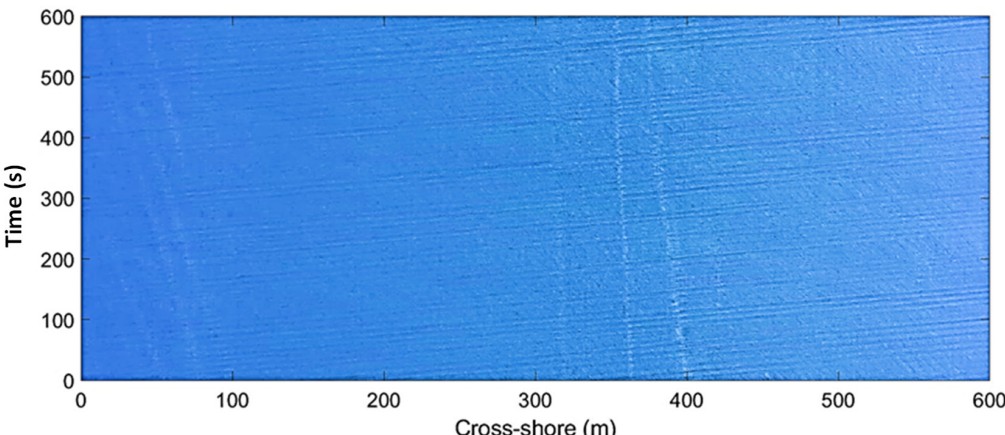

**Figure 12.** Example of the 10-min Timestack generated by the camera on 1 December 2020 (12:50–13:00).

Figure 13 shows the results obtained for both days denoting the agreement between measured and estimated depths along the profile. The estimations practically overlap, allowing to identify the presence of the sandbar and its crest with errors of about 0.25 m. In most of the profile, the errors are below 0.8 m, except for the range $x = (290, 440)$ m. In this case, we confirm the inability of the wave to completely adapt to sudden bottom variations, increasing the errors. It is also noticed that such differences decrease after the passage of the waves through the bar trough, which is in accordance with the patterns identified in the synthetic data case studies. Further offshore of the bar, there is a gradual increase in the error, which might be associated with the spatial resolution to far-field zones of the projected area, but it is also within the range of differences observed for the synthetic cases. The expected similarity of the estimated results for the two consecutive days shows the consistency of the methodology.

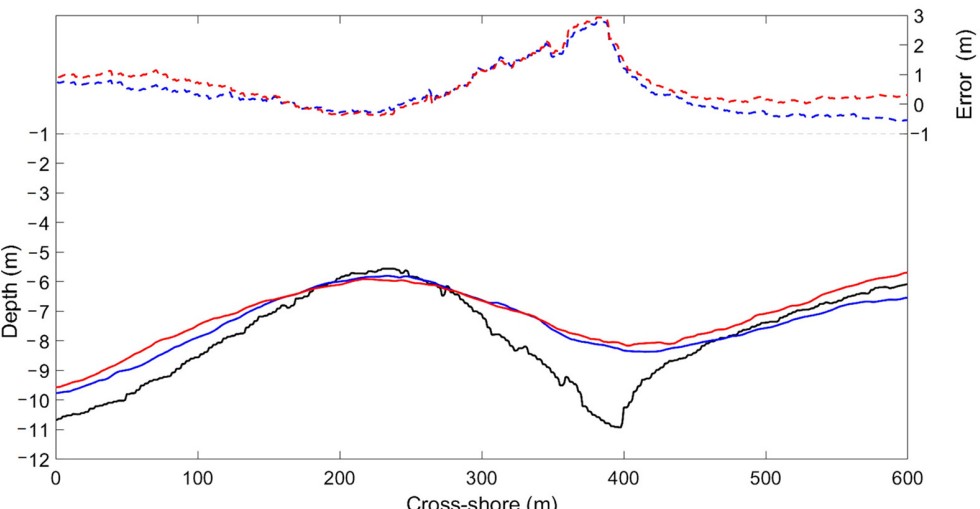

**Figure 13.** Measured bathymetry profile (black line) versus estimated water depths on 1 December 2020 (blue) and on 2 December 2020 (red) and respective errors.

## 4. Discussion

The simulations carried out in this work reveal that the use of a sequence of static images, as obtained from video remote sensing systems, allow to retrieve bathymetric data with great precision ($<O(1)$ m) if no sudden bottom changes appear. Commonly, both optical [37] and radar [38] satellite techniques based on wave characteristics, use a wave frequency estimated offshore and consider it to be constant when the wave approaches the coast. This leads to assume a constant value of $T$, when applying the dispersion relation to retrieve depths [11]. However, if a wide dispersion of wave period values around $T$ exists, large discrepancies between expected and estimated results appear. The improved methodology of Santos et al. [25] for a sequence of images, using the wavelet transform to detect both wavelength and wave period, clearly improves the predictions, with quite small-scale bathymetric errors either for longshore sandbars with different configurations or more complex bathymetric features such as headlands. In addition, the adoption of variable values of $T$ that can be extracted from video imagery does not need to use subdomains or filters as required by Santos et al. [25] when analyzing single static images. Conceptually, the novelty of this method is that the computation of both parameters is intrinsically deriving the wave celerity from the wavelet analysis.

The performance of the method depends on the beach profile depths and the morphology of the bed features. In general, the errors tend to increase offshore, especially for smaller wave periods. For the lower wave period simulations, the interference of wave orbital motions with the bottom is expected to be small or inexistent at deeper waters. In shallow waters, high spatial gradients of the wavelength are expected to occur in such cases, and

the use of several images, identifying wave period variations, enables to estimate depths with greater precision. Despite the complexities of the bathymetric profiles, the differences are not significant. In some places, errors slightly greater than 1 m can be found, but they happen in the presence of sudden morphological variations, so that the waves have no time to completely adapt to the bottom profile. In general, within the region of intermediate and shallow waters, the results between the different wave patterns demonstrate that a better description of sudden changes can be obtained for lower values of $T_p$. The shorter the wave period, the shorter the local wavelength. Therefore, more easily the waves adapt to the underlying seafloor, since there is not a very large variation in depth along the estimated wavelengths. Nevertheless, to extend the prediction to greater depths, longer wave periods are more suitable. This dependency on the wave period was also noted by Thuan et al. [24] by associating error estimates in deep waters with physical limits, since there is a loss of celerity-bathymetry relationship in deep water.

The wavelength and period estimations can also depend on the FUNWAVE spatial and temporal resolutions. The simulations are carried out in specified discrete temporal and spatial domains, and both wavelength and period estimations would be improved if lower values of $\Delta t$ and $\Delta x$ than 1 s and 1 m were adopted. As demonstrated by Pereira et al. [11], when the linear dispersion relation is applied for greater depths, the expected accuracy of the computed depth estimations is lower, depending on wavelength and wave period uncertainties. However, the numerical model was already run on a computing cluster where up to 24 processors were used and the computational cost would be much higher to further reduce the values of $\Delta t$ and $\Delta x$. Despite this, one observes depth differences always bellow 1 m, accurately identifying nearshore morphological singularities as longshore.

When applying the methodology to a real case, the results corroborate what was observed for the synthetic cases. An analysis of two consecutive days allowed to derive consistent depth estimates. The atmospheric and maritime conditions existing on the video images capture, correspond to sunny days, without wind and without wave breaking in the area of interest. Swell waves were observed of around $T = 10$ s and $H_s = 1.2$ m. Challenges related to depth inversion methodologies are to be expected under extreme environmental conditions such as, for example, reduced visibility, large surf zones, strong winds or messy sea states. In the future, it will be interesting to carry out tests on other atmospheric and maritime conditions to evaluate the application limits of this methodology.

## 5. Conclusions

This work proposes a new wavelet-based method for bathymetry retrieval using a sequence of static images of the surface wave field, as obtained from video imagery. This improves Santos et al.'s image processing methodology for single images, by using the wavelet spectral analysis to retrieve wave periods and wavelengths variations.

Synthetic images of the water surface are generated from a numerical Boussinesq type model to simulate the propagation of irregular waves, exploring shoaling and wave refraction processes. The tested conditions lead the waves to break close to the shoreline, excluding surf zone effects from this analysis. The simulations were carried over different beach cross-shore profiles, highlighting the importance to retrieve different wave periods from the sequence of images. This procedure allowed to infer depths for shallow and intermediate waters from the linear dispersion relation with relatively high depth accuracy and for different bottom slopes configurations. It is shown that small-scale bathymetry variations, such as nearshore longshore bars are better reproduced with lower wave peak periods, corresponding to short wavelengths. On the contrary, good depth predictions for deeper depths are obtained with longer wave periods. The application to a shore-based video system confirmed the synthetic data observations, evidencing that developed sandbars in length allow a better detection of the bar crest and its morphology. In addition, errors tend gradually to increase to far-field zones of the projected area.

Promising results to infer variable bathymetries of shallow sandy coasts using this methodology may foster the implementation of new video-based operational systems, sup-

porting the quasi-real time bathymetry where bathymetric surveys are greatly conditioned by favorable maritime and meteorological conditions.

**Author Contributions:** Conceptualization, D.S., T.A. and P.A.S.; methodology, D.S., T.A., P.A.S. and P.B.; software, D.S.; validation, D.S., T.A., P.A.S. and P.B.; formal analysis, D.S., T.A., P.A.S. and P.B.; investigation, D.S., T.A., P.A.S., F.S. and P.B.; resources, D.S.; data curation, D.S. and F.S.; writing—original draft preparation, D.S., T.A., P.A.S. and P.B.; writing—review and editing, D.S., T.A. and P.A.S.; visualization, D.S., T.A. and P.A.S.; supervision, T.A. and P.A.S.; project administration, P.B.; funding acquisition, P.B. All authors have read and agreed to the published version of the manuscript.

**Funding:** This research was funded by Direção Geral da Política do Mar, through project NAVSAFETY of the Fundo Azul program. Thanks are due to FCT/MCTES (PT) for the financial support to CESAM (UIDP/50017/2020 + UIDB/50017/2020 + LA/P/0094/2020), through national funds and to the project Space for Shore funded through EOEP 5 Coastal Erosion Program (ESA/AO/1-9219/18/I-LG).

**Acknowledgments:** The authors gratefully acknowledge Luís Carvalheiro for his help and access to the High-Performance Computational System (ARGUS) at University of Aveiro, enabling running the numerical model on a computing cluster, significantly reducing the time of the simulations.

**Conflicts of Interest:** The authors declare no conflict of interest.

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
