# Peer review of "Nearshore Bathymetry Retrieval from Wave-Based Inversion for Video Imagery"

_remotesensing, doi:10.3390/rs14092155_

Round 1

Reviewer 1 Report

The authors in this paper proposed a method of acquisition using static waveform images. Ultimately, such images can be acquired by video recording, but in this paper they rely on images acquired by a simulation method. Thus, the paper lacks verification of the proposed method on real data. On the other hand, the results of the research are interesting, but they were conducted only at the level of simulation. The paper deals narrowly with the subject of obtaining bathymetry by remote sensing methods, lack of broader discussion, including limitations of the method, which additionally qualifies the paper as a Technical Note

From detailed remarks, Figure 2 does not show items shown in the description (it is exactly Figure 1)

Author Response

Also see the Attached document.

Reviewer #1

The authors in this paper proposed a method of acquisition using static waveform images. Ultimately, such images can be acquired by video recording, but in this paper they rely on images acquired by a simulation method. Thus, the paper lacks verification of the proposed method on real data. On the other hand, the results of the research are interesting, but they were conducted only at the level of simulation. The paper deals narrowly with the subject of obtaining bathymetry by remote sensing methods, lack of broader discussion, including limitations of the method, which additionally qualifies the paper as a Technical Note
R: Thank you for the positive and constructive comments. Following the reviewer’s suggestion, we introduced a new subsection (3.5) where the proposed method is verified with real data. This was performed with images obtained with a recent installed shore-based coastal video station at Figueira da Foz (west coast of Portugal). We recognize that this also helps to expose the relevance of the novel aspects of the presented method, revealing its relevance to the scope of Remote Sensing, qualifying the paper as a Technical Note. In conformity, the abstract, introduction, discussion and conclusions were also adapted. Limitations of the method are presented in the discussion.

From detailed remarks, Figure 2 does not show items shown in the description (it is exactly Figure 1).
R: Thank you very much for your comment. By mistake, there was a repetition of figures, but it has been corrected now.

Reviewer 2 Report

lines 41-42: Although acoustic surveys in the shallowest areas may be economically unreasonable, modern devices allow acquisition in the very shallow waters. Such surveys may be of particular interest when high-resolution bathymetry datasets are needed to acquire. See the example reference: 10.1002/arp.1823.

Moreover, bathymetry LiDAR measurements seem to be a very useful source of high-resolution bathymetry data collection in a short time, but they require specific environmental conditions to be met - see i.e., 10.3390/rs13091750

line 47-48: Explain the difference between remote sensing and conventional surveys

lines 56-63: Consider mentioning the generation of nearshore bathymetry using ICESat-2 and satellite images. Reference: 10.1109/lgrs.2020.2987778

line 89: what do you mean by colors? Please specify

line 138ff: Please change the citation style to the one required by the journal - Author [22].

lines 155-171: Provide the kind of remote sensing equipment used for these bathymetry measurements (SDB / bathy LiDAR / underwater acoustics, etc.)

line 173: I believe that beach cross-shore profiles should look different than it was shown in figure

line 341: Vague. Specify what do you mean by great precision.

line 377: language error 

Author Response

Attached document.

Reviewer 3 Report

Dear Editor and Authors, the manuscript claims to show a depth-inversion bathymetry method for video imagery (see Title), however just a sinthetic numerical example is provided.  Overall, a) remote sensing is not treated. Just a potential sinthetic method is shown and a real case/ application is missing; b) the technical and scientific challenges for remote sensing-based depth-inversion are missing and/or not properly assessed (e.g, image acquisition, image production, image processing, wave signal retrieval, depth-inversion formulas, spatial image resolution, tidal excursion, wind-swell waves, shoaling/breaking waves, ecc.); c) the methods has been already presented by the authors in a previous recent publication [18], and this manuscript just shows small improvements, similar analysis with slightly different hydrodynamics. Large parts of the text recall the previous manuscript.

For all the above reasons, I am not sure the paper is suitable for Remote Sensing journal. I propose Major review, upon requesting the authors i) to focus the manuscripts on remote sensing application more than on hydrodynamic simulation, and ii) to provide at least an example of a real remote sensing application of the proposed method. 

51-70 the Introduction and literature review on image-based depth inversion is very superficial and must be significantly improved for Remote Sensing journal.  First, it is necessary to divide the analysis for different sensors, since type and properties of images much vary  (e.g, type of satellites, X-band radar, video monitoring). Second, different techniques have been developed and used based on sensors and images properties, and these must be clearly indicated (e.g., not only wavelet have been used). Third, it is necessary to, at least, include bathymetry domains provided by different sensors, as well as spatial and temporal resolution, along with numerical accuracy assessed by previous works. All these aspects much differ and influence the applicability.  

If video monitoring is the main topic, all relevant works developed in the last decades are missing, starting from the pioneering Stockdon et al. (2000) to the most recents ones (cBathy, Simarro et al.2019, Thuan et al. 2019 ecc.). Many others can be found in the references therein.

71-72 even Santos et al. [18] does not show image processing algorithm. It is a sinthetic signal analysis

Discussion must also focus on remote sensing applications more than on hydrodynamics. It is missing a comparison with existing techniques. Whether you would apply to satellite or video imagery, advantages and disadvantages of your technique must be commented and compared with the existing ones, after a proper application to field data.

Conclusions. Please, methods and observations must be brief, try to focus more on actual numerical results. 

Author Response

Reviewer #3
Dear Editor and Authors, the manuscript claims to show a depth-inversion bathymetry method for video imagery (see Title), however just a sinthetic numerical example is provided. Overall, a) remote sensing is not treated. Just a potential sinthetic method is shown and a real case/application is missing; b) the technical and scientific challenges for remote sensing-based depthinversion are missing and/or not properly assessed (e.g, image acquisition, image production, image processing, wave signal retrieval, depth-inversion formulas, spatial image resolution, tidal excursion, wind-swell waves, shoaling/breaking waves, ecc.); c) the methods has been already presented by the authors in a previous recent publication [18], and this manuscript just shows small improvements, similar analysis with slightly different hydrodynamics. Large parts of the text recall the previous manuscript.

For all the above reasons, I am not sure the paper is suitable for Remote Sensing journal. I propose Major review, upon requesting the authors i) to focus the manuscripts on remote sensing application more than on hydrodynamic simulation, and ii) to provide at least an example of a real remote sensing application of the proposed method.
R: Thank you for the positive and constructive comments. Following the reviewer’s suggestion several changes were made and we introduced a new subsection (3.5) where the proposed method is verified with real data. This was performed with images obtained with a recent installed shore-based coastal video station at Figueira da Foz (west coast of Portugal). We recognize that this also helps to expose the relevance of the novel aspects of the presented method, revealing its relevance to the scope of Remote Sensing, qualifying the paper as a Technical Note. In conformity, the abstract, introduction, discussion and conclusions were also adapted.

51-70 the Introduction and literature review on image-based depth inversion is very superficial and must be significantly improved for Remote Sensing journal. First, it is necessary to divide the analysis for different sensors, since type and properties of images much vary (e.g, type of satellites, X-band radar, video monitoring). Second, different techniques have been developed and used based on sensors and images properties, and these must be clearly indicated (e.g., not only wavelet have been used). Third, it is necessary to, at least, include bathymetry domains provided by different sensors, as well as spatial and temporal resolution, along with numerical accuracy assessed by previous works. All these aspects much differ and influence the applicability.

If video monitoring is the main topic, all relevant works developed in the last decades are missing, starting from the pioneering Stockdon et al. (2000) to the most recents ones (cBathy, Simarro et al.2019, Thuan et al. 2019 ecc.). Many others can be found in the references therein.
R: Following the reviewer’s suggestion the Introduction and literature review was improved by adding new references given emphasis to video monitoring since is the main topic. We thank the reviewer’s comment, and we believe this improved the quality of this paper.

71-72 even Santos et al. [18] does not show image processing algorithm. It is a sinthetic signal analysis
R: After these lines, we inserted a new reference in the introduction concerning an application of Santos et al. (2020) methodology, providing an example where bathymetry was derived along 220 km of the high energetic Portuguese west coast from satellite images (Santos et al., 2022).

Discussion must also focus on remote sensing applications more than on hydrodynamics. It is missing a comparison with existing techniques. Whether you would apply to satellite or video imagery, advantages and disadvantages of your technique must be commented and compared with the existing ones, after a proper application to field data.
R: As mentioned before, the discussion was changed taking into account the results of the case study (Figueira da Foz). It was also highlighted that, conceptually, the novelty of this method is that the computation of both parameters is intrinsically deriving the wave celerity from the wavelet analysis. The adoption of variable values of T can be extracted from video imagery. We also made reference to Thuan et al.’s (2019) work regarding the dependency on the wave period that corroborates our findings.

Conclusions. Please, methods and observations must be brief, try to focus more on actual numerical results.
R: In conformity, this section was shortened.

Reviewer 4 Report

Thank you for the opportunity to review the paper entitled Nearshore bathymetry retrieval from wave-based inversion for video imagery by Diogo Santos, Tiago Abreu, Paulo A. Silva and Paulo Baptista. The paper is well written and organized. I was also very interested in the paper Estimation of Coastal Bathymetry Using Wavelets by the same authors published in 2020 (J. Mar. Sci. Eng. 2020, 8(10), 772; https://doi.org/10.3390/jmse8100772 ). This first published paper is more methodology oriented, and it is applied to only one site. The new paper here under review applied this method on 3 different sites with different morphologies and explores the advantage of using different discrete wave periods Tp of 6, 10 and 14 s when taking advantage of time series images extracted from video. The major difference between the 20202 paper and this one is mainly in using multiple discrete Tp values to increase the modeled bathy accuracy. Taking advantage of a time series of images extracted from video is already explored and in use for cbathy models using video from an Argus camera for example.

From this paper it is not clear what the ultimate goal of the authors is. One aim could be to come up with an automated algorithm to either process the data with different Tp discrete values, or maybe continuous values over a discrete interval (e.g. 6 to 14 s) and integrate the results where they are the most accurate for a final estimate. Or maybe the goal was to just demonstrate that different Tp value runs give more accurate depth estimates for certain depth ranges. If it is the first aim, then the paper needs a little push forward to demonstrate this integration. If it is the second, then it is an interesting result, but maybe not enough to have a separate paper, but worthy of a conference presentation for example. And this is also because the section 3.4. Wave refraction influence on depth estimation is not fundamentally different from the section presented in the 2020 paper. In the 2020 paper the parameters for oblique waves against a headland were H=2m, T=10sm and angle=30 degrees and for this paper they are: H=1m, t=10s, and angle 15 degrees.

Because of all these comments I would recommend publication with major revisions to make this paper more different than the paper from 2020, and not just a normal extension of work already published.

Author Response

Please all see the Attached document.

Reviewer #4
Thank you for the opportunity to review the paper entitled Nearshore bathymetry retrieval from wave-based inversion for video imagery by Diogo Santos, Tiago Abreu, Paulo A. Silva and Paulo Baptista. The paper is well written and organized. I was also very interested in the paper Estimation of Coastal Bathymetry Using Wavelets by the same authors published in 2020 (J. Mar. Sci. Eng. 2020, 8(10), 772; https://doi.org/10.3390/jmse8100772 ). This first published paper is more methodology oriented, and it is applied to only one site. The new paper here under review applied this method on 3 different sites with different morphologies and explores the advantage of using different discrete wave periods Tp of 6, 10 and 14 s when taking advantage of time series images extracted from video. The major difference between the 20202 paper and this one is mainly in using multiple discrete Tp values to increase the modeled bathy accuracy. Taking advantage of a time series of images extracted from video is already explored and in use for cbathy models using video from an Argus camera for example.

From this paper it is not clear what the ultimate goal of the authors is. One aim could be to come up with an automated algorithm to either process the data with different Tp discrete values, or maybe continuous values over a discrete interval (e.g. 6 to 14 s) and integrate the results where they are the most accurate for a final estimate. Or maybe the goal was to just demonstrate that different Tp value runs give more accurate depth estimates for certain depth ranges. If it is the first aim, then the paper needs a little push forward to demonstrate this integration. If it is the second, then it is an interesting result, but maybe not enough to have a separate paper, but worthy of a conference presentation for example. And this is also because the section 3.4. Wave refraction influence on depth estimation is not fundamentally different from the section presented in the 2020 paper. In the 2020 paper the parameters for oblique waves against a headland were H=2m, T=10sm and angle=30 degrees and for this paper they are: H=1m, t=10s, and angle 15 degrees.

Because of all these comments I would recommend publication with major revisions to make this paper more different than the paper from 2020, and not just a normal extension of work already published.
R: Thank you for the positive and constructive comments. In this new revised version we tried to clarify many aspects of this work. The consideration of both wave periods and wavelength variations along the wave propagation is highlighted in Figure 7, showing the improvements regarding Santos et al.’s (2020) methodology illustrated in Figure 5. The application to satellite images (Santos et al., 2022), where the bathymetry was derived along 220 km of Portuguese west coast only considered wavelength variations and, consequently, required the use of filters and subdomains. No filters and subdomains are applied now. Conceptually, the novelty of this method is that the computation of both parameters is intrinsically deriving the wave celerity from the wavelet analysis. This is now clearly stated in the manuscript and this is possible with video imagery. Moreover, the introduction of a new subsection (3.5) where the proposed method is verified with real data also validates this work. This was performed with images obtained with a recent installed shore-based coastal video station at Figueira da Foz (Portugal). We recognize that this also helps to expose the relevance of the novel aspects of the presented method, revealing its relevance to the scope of Remote Sensing, qualifying the paper as a Technical Note. In conformity, the abstract, introduction, discussion and conclusions were also adapted.

Round 2

Reviewer 1 Report

I am satisfied with the amendments. 

Reviewer 2 Report

Thank you. My comments were addressed properly.

Reviewer 3 Report

Dear Editor and Authors, the revised version of the paper shows some very small improvements, and many of my requirements were not satisfied. The main issue is that the paper does not focus on remote sensing application, which is the interest of journal readers. Therefore, the paper is out of scope for Remote Sensing.

The core of the paper still focus on numerical simulation, suggesting that the paper should be moved to a more suitable journal. Besides, results and conclusions are not coherent with the main aim of paper, and overall the research methods must be revised.

From the remote sensing application point of view, it looks an early stage research. I do not question much the technique itself, but the paper and the research must be much improved before being considered suitable for publication. It must focus on real data and remote sensing application: after an in-depth review of state-of-art, workflow and the several steps required by the technique must be listed, and different cases must also be shown (including issues encountered and low quality assessments).

I list in details here below, in the hope to help authors for a better manuscript.

1- Remote sensing and video monitoring. Introduction does not report coastal video monitoring (30 years) and video depth-inversion (20 years) state-of-art. Authors ignored my requirement of improving the state-of-art: the proposal for a "new" method must also be coupled to a detailed revision of existing techniques. In literature, plenty of different techniques have been previously proposed for video depth-inversion and/or wave celerity estimation (Stockdon, Yoo, Liu, Holman, Zikra, Misra, Almar, Hampson, Simarro, Catalan, Haller, and others) , and many others evaluated their application (Abessolo, Tissier, Thuan, Bergsma, and many others). Neglecting all video properties and state-of-art lead to a poor and not interesting Discussion, where comparison with other works is missing. Finally, missing the state-of-art, it is not clear what the proposed technique is capable of advancing and/or aimed at resolving in comparison to other techniques. Wavelet analysis for retrieving wave parameters from synthetic images has already been shown in the previous authors publication, therefore should be applied on real data and more in-depth analysis must be done.

Overall, reference list (33 papers) is very limited for a research paper dealing with the depth-inversion topic, also considering that just six of those are related to video depth-inversion. It is not mentioned any of the final assessments obtained by previous techniques, pro and cons ecc..

2- Real-case video application. Even though a real-case is shown at the end of Results, the methodological explanation of the real-case is very superficial and does not provide enough and/or useful information to readers. The final results of depth-inversion are very good, however it is difficult to evaluate their consistency if the whole video processing and depth-inversion process are neglected in the explanation, as I previously requested (e.g, image acquisition, image production, image processing, wave signal retrieval, wave orientation, depth-inversion formulas, spatial image resolution, tidal excursion, wind-swell waves, shoaling/breaking waves, outliers, ecc.). The new 359-399 barely mentioned the whole process, providing a very superficial explanation. The new section does not report any of the remote-sensing based process used to deal with the several depth-inversion challenges. It is not clear if one single Timestack was used, or if multiple images were considered. In both cases, it is necessary to report when and if same results are obtained by other single images, or how many images were used to reach acceptable results (eventually, how technique deals with outliers, low quality images ecc.). Finally, the technique must be shown applied and tested on different hydrodynamics, to understand likely limitations and applicability.

3- Over-simplified hydrodynamic case. It is clear that to explore the potential of proposed technique in retrieving wave celerity, even from synthetic images, must be shown in different tidal and wave height scenarios, as also mentioned by authors in 454-455. Considering just a single wave height (1 m) and avoiding tidal excursion, seems to be a choice to avoid wave breaking and therefore simplifying the whole analysis. In fact, the paper does not show or cite any limitation and/or issues encountered by the technique. Therefore, the analysis is very limited and does not properly explore the potentiality and/or limitations of the technique. This also because most of video-based depth-inversion techniques, obviously focused on nearshore due to logistical limitation of video monitoring, shown to be less performing in the outer surf zone (apparent breaking wave height acceleration), and/or during higher breaking wave height (stronger non-linear wave effect), and/or during very low wave height (small intensity wave signal).

Finally, the whole simulation of depth-inversion, based on Boussinesq wave model, basically tested the goodness of depth-inversion of a Boussinesq wave field by the linear wave theory. Several works, based on numerical models and simulations, have already tested that (and must be also present in the literature review).

4- Conclusions. Even from a superficial revision of the state-of-art, authors have already found that similar observations and conclusions have been already pointed out and presented by other authors (e.g., Thuan, but not only). The main Conclusions obtained from simulation refer the importance of wave period in depth-inversion, but from the analysis, this looks true out of the domain of video-based technique: it was not necessary to simulate waves till 3 km from the shore, since images from video systems worldwide can provide images at most 600 m from the shore (as shown in real-case, and can be seen in literature). The numerical simulation, analysis and Conclusions are therefore not in line (and not of interest) with the main objectives of the paper, which aims at exploring the feasibility of the technique for video-based technique.

These suggest that more robust research must be carried out to improve the novelty of the work and the final conclusions. I have already pointed out in the previous review that the spatial domain of interest must be defined and set properly. If Authors wish to improve numerical simulations, I suggest at least to show the differences of wave parameters retrieved by wavelet and by others technique previously presented by other works, even though I do not believe there will be substantial differences using synthetic images.